# Safety and Efficacy of Atogepant for the Preventive Treatment of Migraines in Adults: A Systematic Review and Meta-Analysis

**DOI:** 10.3390/jcm13226713

**Published:** 2024-11-08

**Authors:** Abdulrahim Saleh Alrasheed, Taif Mansour Almaqboul, Reem Ali Alshamrani, Noor Mohammad AlMohish, Majed Mohammad Alabdali

**Affiliations:** 1College of Medicine, King Faisal University, AlAhsa 31982, Saudi Arabia; 2College of Medicine, Batterjee Medical College, Jeddah 21442, Saudi Arabia; 150341.taif@bmc.edu.sa; 3College of Medicine, Taif University, Taif 21944, Saudi Arabia; reemalshamrani-@hotmail.com; 4Neurology Department, King Fahad Hospital of the University, Imam Abdulrahman Bin Faisal University, Khobar 34445, Saudi Arabia; nmalmohish@iau.edu.sa; 5Neurology Department, College of Medicine, Imam Abdulrahman Bin Faisal University, Khobar 34445, Saudi Arabia; mmalabdali@iau.edu.sa

**Keywords:** atogepant, CGRP receptor antagonist, preventive treatment, episodic migraine

## Abstract

**Background:** Migraine is a common neurological condition marked by unilateral recurrent pulsating headaches, often associated with systemic signs and symptoms. Recently, calcitonin gene-related peptide (CGRP) antagonists, including atogepant, an oral CGRP receptor antagonist, have emerged as effective and safe treatments. The current study sought to assess the efficacy and safety of atogepant for preventing episodic migraines in adults. **Methods:** A comprehensive search, following PRISMA guidelines, was conducted using PubMed, Web of Science, and Cochrane Library to identify randomized, double-blind, placebo-controlled trials published up to June 2024. **Results:** The studies included adult participants with episodic migraine treated with atogepant. The primary outcomes assessed were changes in mean monthly migraine days (MMDs) and monthly headache days (MHDs) over 12 weeks. Secondary outcomes included reduction in acute medication use, 50% responder rates, and adverse events. A meta-analysis using a random-effects model was performed to evaluate efficacy and safety. Six trials with 4569 participants were included. Atogepant significantly reduced mean monthly migraine days (MMDs) and monthly headache days (MHDs) compared to placebo at all doses (10 mg, 30 mg, 60 mg), with the 60 mg dose showing the greatest reduction (mean difference: −1.48 days, *p* < 0.001). Significant reductions in acute medication use and improved 50% responder rates were also observed for all doses. The safety profile of atogepant was favorable, with common adverse events being mild to moderate, such as constipation and nausea. There were no significant differences in serious adverse events between the atogepant and placebo groups. **Conclusions:** Atogepant is an effective and well-tolerated option for preventing episodic migraines, showing significant reductions in migraine frequency and acute medication use. However, further studies are necessary to assess its long-term safety and efficacy, especially at higher doses, and to investigate its potential role in personalized treatment strategies for migraine prevention.

## 1. Introduction

Migraine is a prevalent and debilitating neurological condition marked by unilateral recurrent pulsating moderate to severe attacks of headache, often associated with systemic symptoms such as vomiting, nausea, and sensitivity to sound and light [1]. It affects approximately 12% of the global population, with a higher incidence in women compared to men [2,3]. Migraines disproportionately affect individuals during their most productive years, typically between the ages of 18 and 44 [3]. The recurrent nature of migraines leads to frequent absenteeism and presenteeism, severely impacting professional and academic performance [4]. Moreover, the unpredictable onset of migraines contributes to anxiety and depression [5]. Migraine significantly impacts quality of life, leading to substantial personal, social, and economic burdens [6].

Migraine can be classified into two major types according to the International Classification of Headache Disorders (ICHD-3): episodic and chronic. The episodic form is characterized by headache attacks occurring on less than 15 days per month, whereas the chronic form involves headaches on 15 or more days per month, persisting for more than 3 months, with the diagnostic criteria including at least 8 days per month of migraine headaches. The distinction between these types is important for treatment and management strategies [7].

Treatment for migraine includes both acute and preventive approaches [8]. Acute treatments, which include triptans, non-steroidal anti-inflammatory drugs (NSAIDs), and antiemetics, are intended to relieve symptoms during an attack. Preventive treatments, including botulinum toxin A, antidepressants, anticonvulsants, and antihypertensive drugs, are used to lower the incidence, severity, and duration of migraines, thereby improving the overall quality of life for sufferers [8,9]. Approximately 50–60% of patients benefit from preventive treatments, highlighting their importance in migraine management [10,11]. However, these were traditional treatments for migraine, and they are non-specific migraine treatments with limited degrees of effectiveness, prompting the need for more effective and tolerable therapeutic options [12,13].

The advent of calcitonin gene-related peptide (CGRP) receptor antagonists represents a significant advancement in the management of migraine [14]. CGRP plays a crucial role in migraine pathophysiology by modulating pain pathways and vascular functions. CGRP is released during migraine attacks and contributes to the inflammation and dilation of cerebral blood vessels. The development of CGRP antagonists offers a targeted approach to migraine treatment, providing relief with fewer side effects compared to traditional therapies [15].

For the prevention of migraine, there are several monoclonal antibodies acting on the CGRP pathway; erenumab targets the CGRP receptor, while eptinezumab, fremanezumab, and galcanezumab block the CGRP ligand. Rimegepant and ubrogepant, two oral CGRP receptor antagonists, are licensed for the management of migraine attacks. Rimegepant also gained approval for the prevention of migraines in adults in 2021, making it the only medication that may be used for both acute and preventive treatment of migraines [16]. Although these monoclonal antibodies (MAbs) have been used for the prevention of migraine attacks, patients experienced some discomfort from the subcutaneous or intravenous delivery of these medications [12]. In contrast to preventive monoclonal antibodies, CGRP receptor antagonists (gepants), are primarily administered as pills, nasal sprays, and orally disintegrating tablets (ODT) [17].

Atogepant stands out as one of the oral drugs established for the preventive treatment of episodic migraine. It was accepted by the FDA on 15 September 2021, and is available in doses of 10 mg, 30 mg, and 60 mg [9].

Approximated to additional CGRP receptor antagonists like rimegepant, atogepant, an oral CGRP receptor antagonist authorized for the preventive treatment of episodic migraine, has several significant benefits [18,19]. Rimegepant is certified for both acute and preventive therapy; however, because it is dosed every other day for prevention, adherence issues may arise. Contrarily, atogepant is only meant to be taken once daily, making it easier to follow and more convenient for patients [19]. Furthermore, atogepant has a well-established track record of lowering monthly migraine days (MMDs), with notable reductions in MMDs and a positive safety record. This once-a-day oral medication provides an opportune, secure, and efficient preventive outcome [18]. It is especially appropriate for individuals who would choose a daily dosage regimen without the intrusiveness of injectables [19]. The current study sought to assess the efficacy and safety of atogepant for preventing episodic migraines in adults.

## 2. Methods

This systematic review and meta-analysis adhered to the Preferred Reporting Items for Systematic Reviews and Meta-Analyses (PRISMA) guidelines. Prospectively, we registered the study protocol in the International Prospective Register of Systematic Reviews (PROSPERO) (registration number: CRD42024556275).

### 2.1. Search Strategy

We systematically searched the PubMed, Cochrane Library, and Web of Science databases through June 2024. The search strategy used a combination of Medical Subject Headings (MeSH) and the following Boolean operators: “Migraine” OR “Episodic Migraine” OR “Recurrent Migraine” OR “Refractory Migraine” OR “Headache” OR “Recurrent Headache” OR “Refractory Headache” AND “Efficacy” OR “Safety” OR “Tolerability” OR “Outcome” OR “Findings” OR “Impact” AND “Atogepant” OR “Preventive Treatment” OR “Preventive Therapy” OR “Prevention” AND “Random” OR “Placebo” OR “Trial” OR “Group”. The retrieved study reference lists were revised to identify other relevant articles.

### 2.2. Eligibility Criteria

The studies included in this review were randomized, placebo-controlled trials involving participants aged ≥18 years with a history of migraine for at least twelve months, with onset before age 50. Participants needed to experience 4–14 monthly migraine days (MMDs) during the three months prior to screening and record this data in an electronic diary during a 28-day baseline period. Only English-language papers were considered. Exclusion criteria included studies that did not report outcomes of interest, review articles, and case reports.

### 2.3. Outcome Measures

The evaluated primary efficacy outcomes included changes from baseline in the mean number of monthly migraine days (MMDs), monthly headache days (MHDs), and the number of acute-medications-use days per month. Additionally, the percentage of participants who experienced at least a 50% reduction in migraine days per month during the double-blind treatment phase was assessed.

Secondary efficacy outcomes involved evaluating changes from baseline in the average number of acute-medications-use days. Safety and tolerability outcomes included rates of adverse events (AEs), withdrawals due to AEs, and serious adverse events (SAEs). Key safety measures were treatment emergent adverse events (TEAEs).

### 2.4. Study Selection, Data Extraction, and Quality Assessment

We used Rayyan Software (Version 1.5.0, Qatar Computing Research Institute, Doha, Qatar) to manage electronic database search results for selection, screening, and duplicate removal. Titles and abstracts were screened by two independent reviewers, with any disagreements resolved by involving a third reviewer.

Data extraction was conducted by two independent reviewers, focusing on study characteristics, participant demographics, intervention specifics, and outcomes. Any disagreements were resolved through discussion. Extracted data included author details, publication year, journal name, country, study design, and key trial elements such as randomization, blinding, treatment periods, and atogepant doses (10, 30, and 60 mg once daily). Screening, randomization, and follow-up periods were noted, alongside inclusion and exclusion criteria. Participant demographics covered age, gender, race, and ethnicity. Outcome measures, along with the total number of participants, group distributions, analysis populations, and safety metrics, were documented.

The risk of bias was independently assessed by two reviewers using the Cochrane Risk of Bias 2 (RoB 2) tool via Review Manager 5.4 software (The Cochrane Collaboration, London, UK), with any disagreements resolved through discussion or consultation with a third reviewer [20,21]. The quality assessment of the included studies was conducted utilizing the Revised Cochrane Risk of Bias Assessment Tool 2. The evidence certainty for each outcome was evaluated using the GRADE approach, which considered factors such as the robustness of the data, potential selective reporting, and other bias sources. Evidence was downgraded for serious or very serious concerns [22].

### 2.5. Statistical Analysis

Heterogeneity among the trials was evaluated by the funnel plot visual examination, the chi-squared test, and I^2^ statistics. A fixed-effect model was planned for use when heterogeneity was not significant (*p* > 0.05) [23,24]. However, the decision between a fixed-effect or random-effects model was primarily based on the I^2^ value: a fixed-effect model was planned to be applied when I^2^ was less than 40%, while a random-effects model was utilized for I^2^ values of 40% or greater. The measures of association between treatment and continuous or dichotomous outcomes were the mean difference (MD) and risk ratio (RR), both reported alongside 95% confidence intervals (CIs).

During the double-blind treatment period, all efficacy analyses were carried out on the modified intention-to-treat population, encompassing all randomly allocated participants who had a minimum of one dose of atogepant, assessable baseline electronic recorded data, and a minimum of one evaluable post-baseline 4-week period of electronic recorded diary. All individuals who received at least one dosage of the study drug were included in the safety and tolerability evaluations. Two-sided *p*-values were reported, and *p* < 0.05 was considered significant. The data were analyzed using Review Manager software version 5.4.

## 3. Results

### 3.1. Search Results

A flowchart illustrating the study selection process is shown in (Figure 1). Our search identified 71 records. These included 35 duplicate articles that were removed, leaving 36 unique articles for screening by title and abstract. Out of the 36 articles screened, 18 were excluded as they did not meet the eligibility criteria. The remaining 18 full-text articles were reviewed for more detailed evaluation and 12 articles were excluded because 9 of them did not report on relevant episodic migraine preventive therapy and three articles did not report the outcomes of interest. Finally, six studies were included in our study [9,11,25,26,27,28].

### 3.2. Quality Assessment

In the risk-of-bias assessment, three of the six included studies exhibited potential concerns. Specifically, issues related to randomization and missing outcome data were identified in two trials [9,25]. In contrast, the remaining three studies [11,26,28] were determined to have a minimal risk of bias (Figure 2).

### 3.3. Characteristics of the Included Studies

The current review included six studies, where a total of 1231 participants were allocated randomly to placebo, 977 to atogepant 10 mg, 1095 to atogepant 30 mg, and 1266 to atogepant 60 mg, for a total of 4569 participants [9,11,25,26,27,28]. Participants’ ages ranged from 18 to 73 years; 87.2% were females, 12.8% were males, and 81.9% were white. However, no data were provided about the patients’ ethnic origin in any of the studied articles. The mean BMI ranged from 26.2 (5.2) in Tassorelli et al.’s 2024 [28] study to 31.1 ± 7.6 kg/m^2^ in Ailani et al.’s 2021 [11] study. Among studies that provided relevant data, 97.8% of the participants reported current use of acute medications. In terms of migraine frequency, the monthly migraine days (MMD) during the 28-day baseline period ranged from 7 to 14 days. Notably, only Goadsby et al., 2020 [26] assessed the migraine associated with aura, where 21.5% of the participants reported such association. Baseline monthly headache days (MHD) were assessed in four articles [11,25,26,27] and were found to range from 4 to 14 days. Furthermore, only two studies [25,27] assessed the baseline monthly acute-medications-use days, reporting a mean of 6.5 to 6.9 days (Table 1).

### 3.4. Meta-Analysis

The forest plots and meta-analysis were performed using Review Manager software version 5.4. All results were subjected to a random-effects model. Mean differences (MD) were utilized to report continuous outcomes, while risk ratios (RR) were used to represent dichotomous outcomes. Publication bias was visually assessed using a funnel plot, and heterogeneity was quantified using the I^2^ statistic and the Chi^2^ test *p*-value. A 95% confidence interval was applied to all estimates, and results were considered statistically significant if the *p*-value was less than 0.05.

### 3.5. Mean Monthly Migraine Days Change from Baseline

Three studies were included in the analysis, revealing that atogepant significantly outperformed placebo in reducing mean monthly migraine days across various subgroups. The observed mean differences were as follows: −1.16 days for atogepant 10 mg (*p* < 0.001, I^2^ = 0%), −1.15 days for atogepant 30 mg (*p* < 0.001, I^2^ = 36%), and −1.48 days for atogepant 60 mg (*p* = 0.0009, I^2^ = 79%). While no significant heterogeneity was detected in the atogepant 10 mg and 30 mg subgroups, a substantial degree of heterogeneity was present in the atogepant 60 mg subgroup, attributed to random error. The *p*-value for the atogepant 60 mg subgroup was 0.0009, with an I^2^ of 79%. Although the study by Schwedt et al. (2022) was excluded from this analysis due to missing data, its findings align with those of the included studies. Schwedt et al. (2022) reported that atogepant significantly reduced the mean monthly migraine days from baseline across all dosages (10 mg, 30 mg, and 60 mg) over all follow-up periods (1–4 weeks, 5–8 weeks, and 9–12 weeks) [25] (Figure 3).

### 3.6. Mean Monthly Headache Days Change from Baseline

This analysis included three trials, each of which indicated a substantial reduction in mean monthly headache days from baseline throughout a 12-week follow-up period for all atogepant dosages. None of the subgroups exhibited any detectable heterogeneity. Monthly headache days were reduced by 1.40 days for atogepant 10 mg (*p* < 0.001, I^2^ = 0%), −1.44 days for atogepant 30 mg (*p* < 0.001, I^2^ = 0%), and −1.63 days for atogepant 60 mg (*p* < 0.001, I^2^ = 49%). Furthermore, the study conducted by Schwedt et al. (2022) confirms the efficacy of atogepant (10 mg, 30 mg, and 60 mg) in dramatically lowering the mean number of monthly headache days from baseline over a 4-week period [25] (Figure 4).

### 3.7. Acute Medication Use Days Change from Baseline

Over the 12-week follow-up period, atogepant (10 mg, 30 mg, and 60 mg) significantly reduced the requirement for acute medicines. There was no noticeable heterogeneity among the subgroups. Atogepant’s mean differences at 10 mg, 30 mg, and 60 mg were −1.30, −1.40, and less than 0.001, I^2^ = 0%, and −1.58, with a *p*-value less than 0.001 and I^2^ = 65%, respectively. Schwedt et al. (2022) found that atogepant considerably reduced the requirement for acute treatment at all doses (10 mg, 30 mg, and 60 mg) and follow-up intervals (1–4 weeks, 5–8 weeks, and 9–12 weeks [25] (Figure 5).

### 3.8. ≥Outcome of 50% Reduction in Monthly Migraine Days

A forest plot was utilized to analyze three of the selected studies, focusing on the outcome of a 50% reduction in monthly migraine days. The analysis demonstrated that, compared to a placebo, atogepant at doses of 10 mg, 30 mg, and 60 mg resulted in a reduction of more than 50% in monthly migraine days over the 12-week follow-up period. Significant heterogeneity was noted within the 30 mg and 60 mg subgroups. The relative risks (RR) for the different dosages of atogepant were as follows: 10 mg (RR = −1.66; *p* = 0.003; I^2^ = 65%), 30 mg (RR = 1.63; *p* = 0.02; I^2^ = 85%), and 60 mg (RR = 1.94; *p* = 0.003; I^2^ = 87%) (Figure 6).

### 3.9. Adverse Events

According to Goadsby et al. (2020), the incidence of treatment-emergent adverse events (TEAEs) increased with higher doses of atogepant, rising from 18% in the 10 mg once-daily subgroup to 26% in the 60 mg twice-daily subgroup, while the placebo group experienced a lower rate of 16% [26]. Nausea was identified as the most common treatment-related TEAE, occurring in 3–6% of the once-daily dose subgroups and 6–9% of the twice-daily dose subgroups, with rates ranging from 3% in the 10 mg once-daily group to 6% in the 60 mg once-daily group, compared to 3% for placebo. Notably, no evidence hepatic injury was reported [26].

Schwedt et al. (2022) reported that the proportion of patients having treatment-emergent adverse events (TEAEs) was consistent across all groups, ranging from 52.2% to 53.7% in the atogepant treatment groups and 56.8% in the placebo group. Although the study was unable to specify the adverse events that caused cessation, it did indicate that 1.8–4.1% of the atogepant groups had such events, compared to 2.7% in the placebo group. A total of 486 out of 902 participants (53.9%) reported adverse events that started or got worse after the first dose of atogepant or placebo until 30 days after the final dose. The occurrence of events was comparable in both groups, and there was no dose relationship observed [25]. The most frequently reported side effects in the study by Ailani et al. (2021) were constipation (6.9% to 7.7% across atogepant dosages) and nausea (4.4% to 6.1% across atogepant doses). Ocular neuritis and asthma were among the serious adverse effects reported in the 10 mg atogepant group [11].

Lipton et al.’s 2022 trial found that the proportion of individuals experiencing TEAEs was similar across all atogepant groups (52.9% in the 10 mg group, 52.2% in the 30 mg group, and 53.7% in the 60 mg group) compared to the placebo group (56.8%) [9].

The most frequently reported TEAEs were constipation (7.7% in the 10 mg group, 7.0% in the 30 mg group, and 6.9% in the 60 mg group) and nausea (5.0% in the 10 mg group, 4.4% in the 30 mg group, and 6.1% in the 60 mg group), compared to 0.5% for constipation and 1.8% for nausea in the placebo group, respectively [9]. However, Lipton et al. (2023) did not describe any specific adverse events in their report [27].

According to a recent study by Tassorelli et al. (2024), 84 (54%) participants in the placebo group reported treatment-emergent adverse events compared to 81 (52%) in the atogepant group. Constipation was the most common TEAE with atogepant, occurring in 10% of cases versus 3% in the placebo group. Tassorelli et al. (2024) discovered that 3% of participants in the atogepant subgroup and 1% in the placebo group experienced significant adverse events, while 2% of individuals in the atogepant group and 1% in the placebo group encountered TEAEs that required treatment cessation [28] (Table 2).

### 3.10. Any Adverse Events

Overall, three investigations were examined in this analysis. Although adverse events related to atogepant treatment did not increase significantly across all dosage categories, there was significant variability in the atogepant 10 mg and 30 mg subgroups. Atogepant 10 mg, 30 mg, and 60 mg had relative risks (RRs) of 1.11 (*p* = 0.57; I^2^ = 85%), 1.08 (*p* = 0.64; I^2^ = 85%), and 1.02 (*p* = 0.78; I^2^ = 29%). In all studies employing atogepant, the most commonly reported adverse events were nausea, constipation, and upper respiratory tract infection. Furthermore, Schwedt et al. (2022) found no difference in the occurrence of these side effects across all groups compared to the placebo [25] (Figure 7).

### 3.11. Serious Adverse Events

The three studies included in this forest plot on serious adverse events demonstrated that atogepant was comparable to a placebo and did not significantly increase the occurrence of such events. No discernible variation was observed across any of the groups. The results for atogepant 10 mg, 30 mg, and 60 mg were as follows: (RR = 1.00, *p* = 1.00, I^2^ = 0%), (RR = −0.63, *p* = 0.58, I^2^ = 0%), and (RR = 0.62, *p* = 0.62, I^2^ = 0%), respectively (Figure 8).

### 3.12. Discontinuation Due to Adverse Events

This analysis incorporated data from five studies and determined that atogepant did not significantly increase the incidence of medication discontinuations due to adverse events in comparison to a placebo. Neither subgroup exhibited any notable heterogeneity. The three dosages of atogepant evaluated were 10 mg (RR = −1.09; *p* = 0.57; I^2^ = 25%), 30 mg (RR = 1.01; *p* = 0.94; I^2^ = 39%), and 60 mg (RR = 0.98; *p* = 0.79; I^2^ = 0%). Schwedt et al. (2022) reported comparable findings, with no evidence of a dose–response relationship [25] (Figure 9).

## 4. Discussion

In our study, we pooled data from six randomized controlled trials (RCTs) involving a total of 4569 patients to evaluate the efficacy and safety of atogepant for episodic migraine prevention. Through the analysis, different atogepant dosages (10 mg, 30 mg, and 60 mg), demonstrated that atogepant significantly reduced both mean monthly migraine days (MMDs) and mean monthly headache days (MHDs) compared to placebo over a 12-week follow-up period. The reduction was consistent across the trials, demonstrating the drug’s efficacy in preventing migraines and the ability to alleviate the overall headache burden. This broader impact is particularly relevant for patients whose migraine-related disability extends beyond the headache itself, affecting overall quality of life [28].

The included trials also highlighted other key efficacy endpoints, such as the reduction in acute medication use, which is a critical factor for patients at risk of medication-overuse headaches (MOH), a common issue among migraine sufferers, which is a crucial finding, as reducing acute medication use not only alleviates symptoms but also mitigates the risk of developing MOH, which can exacerbate the condition [5]. Furthermore, the 50% responder rate analysis, which defined responders as patients achieving a ≥50% reduction in MMDs, showed significant improvements across all atogepant doses. The 10 mg dose demonstrated a relative risk of 1.66 (*p* = 0.007) compared to placebo, while the 30 mg dose showed a relative risk of 1.63 (*p* = 0.02) compared to placebo. The highest efficacy was observed with the 60 mg dose, which demonstrated a relative risk of 1.94 (*p* = 0.003) compared to placebo, indicating that higher doses yield greater benefits in terms of reduction in monthly migraine days.

According to Lipton et al. (2024), the dose–response relationship of atogepant shows that most participants who initially reported improvements continued to experience lasting benefits throughout the treatment period, particularly at higher doses. The 60 mg dose, in particular, led to better initial response rates and continued effectiveness in decreasing monthly migraine days. The data indicate that the highest dose resulted in the most participants achieving and sustaining responses at all levels of MMD reductions (50%, 75%, and 100%). These results endorse atogepant as a feasible choice for preventing episodic migraine, emphasizing the benefits of higher doses for the best treatment results [29].

When it comes to the safety profile, our review confirmed that atogepant has a favorable profile with no significant increase in adverse events compared to placebo. The most commonly reported side effects were mild, including nausea, constipation, and upper respiratory tract infections. Importantly, there was no significant dose–response relationship for adverse events, meaning that increasing the dose did not lead to a proportionate increase in adverse reactions. The absence of serious adverse events across all doses further supports atogepant’s safety, making it a safer alternative to other preventive migraine therapies, such as triptans, which are often associated with cardiovascular risks [30]. This safety profile, combined with its efficacy, positions atogepant as a well-balanced option for the long-term prevention of episodic migraine.

Even though atogepant has a favorable safety profile, with no significant increase in adverse events compared to placebo, there are some reported adverse events. These findings reinforce and expand upon previous studies, establishing atogepant as an effective prophylactic treatment for episodic migraine [12,16,31,32].

The ADVANCE trial offered compelling evidence of rapid onset, with individuals having significant reductions in MMDs as early as the first week, an observation that corresponds with our findings of consistent improvements across all atogepant dosages [26]. Tassorelli et al. (2024) found that implementation of once-daily 60 mg atogepant was safe and well tolerated, and led to a significant and clinically relevant reduction in mean monthly migraine days over the course of 12 weeks when compared to placebo in patients with episodic migraine. Prior to now, two to four classes of traditional oral preventive medications had failed to achieve remission in these patients [28].

Based on estimates from the 2016 Global Burden of Disease Study, migraines rank as the second leading cause of disability-adjusted life-years (DALYs) worldwide, underscoring their substantial impact on global health [33]. Although several treatment regimens have shown efficacy in migraine prevention, there is a significant proportion of migraine patients reported to be frequent visitors of physicians’ clinical practice, highlighting the urgent need to implement an efficacious and well-tolerated drug for migraine prophylaxis [5].

Atogepant, a small-molecule calcitonin gene-related peptide (CGRP) receptor antagonist, is one of the newer preventive treatments developed to manage episodic migraine. CGRP is a neuropeptide that plays a significant role in the pathophysiology of migraine, contributing to inflammation and vasodilation that can lead to headache onset [14]. By blocking the CGRP receptor, atogepant helps to prevent the onset of episodic migraine attacks. Its oral formulation allows for ease of use, particularly for patients who need daily preventive therapy. Unlike other classes of migraine treatment, such as triptans, which have vasoconstrictive properties and are associated with cardiovascular risks, atogepant does not carry such risks [5]. Furthermore, when it comes to efficacy, atogepant is found to be more effective in reducing episodic migraine attacks when compared to the other most common and non-specific migraine prophylactic drugs, such as beta-blockers and amitriptyline, making it a suitable option for a broader range of patients [34].

Studies have assessed the safety of CGRP small-molecule antagonists, called gepants, and found that they are well tolerated overall. For example, ubrogepant, one of the initial gepants authorized by the FDA, has been linked to typical side effects like nausea and drowsiness, but severe side effects are uncommon. Clinical trials show that ubrogepant has similar rates of adverse events to placebo, demonstrating its safety for treating acute migraines. Rimegepant, another type of gepant, demonstrates a comparable safety record, with the majority of negative effects being mild to moderate, such as headaches and dizziness. Both ubrogepant and rimegepant do not have significant cardiovascular risks, which makes them appropriate choices for patients with a cardiovascular history. In general, gepants offer a hopeful safety record while efficiently treating migraine episodes [30].

The American Headache Society (AHS) released a consensus statement advocating the use of CGRP-targeting medicines as a primary option for migraine prevention in addition to prior first-line therapies without requiring a prior attempt and failure of other migraine-preventive medications. The statement emphasizes the long-term tolerability and efficacy of these medications, including atogepant [35]. This conclusion was further supported by the recent 2024 National Institute for Health and Care Excellence (NICE) guidelines, which recommend atogepant as migraine preventive medication in adults who experience a minimum of four migraine days per month, but only after failing a minimum of three preventive medications [36].

Raja et al. (2024) underscore the significance of personalized treatment approaches in the management of migraines. Although all administered doses demonstrated efficacy in decreasing the number of migraine days, the analysis did not reveal a distinct dose–response relationship. Notably, higher doses, especially the 60 mg once-daily regimen, were associated with improvements in functional outcomes and a reduction in the need for acute medications, indicating a more pronounced impact on quality of life. Conversely, the efficacy of twice-daily dosing did not consistently surpass that of once-daily regimens. This suggests that lower doses may be sufficient for certain patients, while those with more severe symptoms might benefit from higher doses [37].

Previous studies, including that conducted by Tao et al. (2022) [12] and Lattanzi et al. (2022) [16] revealed that atogepant is an effective and well-tolerated episodic-migraine-preventive treatment. Furthermore, a notable heterogeneity in the ability to reduce MHDs or MMDs in the 60 mg dose group was observed, which could reflect variability in patient populations, such as demographic differences, supporting the conclusion that a personalized treatment approach may be necessary to tailor the dosage of atogepant based on individual patient factors, such as baseline migraine severity and co-morbidities. However, the findings were largely in alignment with ours; we included a greater number of RCTs and a larger sample size, which strengthened the validity of applying the results to the individual clinical practice.

In their meta-analysis, Hou et al., 2024 [31] discovered that patients who received a daily dose of atogepant 10 mg, 30 mg, or 60 mg experienced a considerably higher decrease in the mean number of migraine days from baseline than those given a placebo. Accordingly, the evaluation determined that atogepant is an effective and generally well-tolerated therapy for adult episodic migraine prophylaxis.

A recent meta-analysis conducted by Lopes et al., 2024 [32], further supported the effectiveness and tolerability of atogepant for migraine prevention, including episodic or chronic migraine, when compared to placebo. They noticed that in terms of the monthly reduction of migraine or headache days, the overall impact estimate of atogepant was much greater than that of placebo.

Although our findings back up these recent meta-analyses’ findings [31,32], our meta-analysis offers a more focused assessment by exclusively including episodic migraine patients. This specificity enhances the accuracy of the findings for this patient population. The inclusion of both episodic and chronic migraine populations may have diluted the efficacy results for episodic migraine. By focusing solely on episodic migraine, we provide more reliable and precise insights into the efficacy of atogepant for this group, considering the different pathophysiological features of chronic when compared to episodic migraines.

### Strengths and Limitations

Compared to the current literature, our study provides a comprehensive and focused assessment by including only studies that specifically evaluate the safety and efficacy of atogepant in episodic migraine patients. However, several limitations should be noted: Firstly, the meta-analysis incorporated a relatively small number of randomized controlled trials (RCTs) with a short follow-up period of 12 weeks. These factors may constrain the applicability of the results to wider contexts. This brief period may not be sufficient for assessing atogepant’s long-term safety and efficacy, especially for chronic conditions such as migraine that require ongoing treatment. One important limitation is the absence of extended safety evaluations, making it harder to predict potential late-onset side effects and the lasting effectiveness of medical results.

Furthermore, the results may not be as broadly applicable, as they may be due to different demographic considerations, since all the research efforts were carried out in Western nations. Finally, the lack of sufficient reported outcomes made it difficult to perform a meta-analysis of adverse events (AEs).

## 5. Conclusions

In patients with episodic migraine, atogepant has shown notable effectiveness in lowering MMDs, MHDs, and acute drug use while raising the 50% responder rate. The need for individualized treatment plans is highlighted by the observed variability in the higher-dose groups. To verify atogepant’s long-term effectiveness and safety, especially in larger patient groups, and to evaluate its possible function in conjunction with other treatments for the best migraine care, more extensive clinical trials are required. Future research should consider geographical and demographic differences across populations to help shape a new era of personalized treatment approaches.

## Figures and Tables

**Figure 1 jcm-13-06713-f001:**
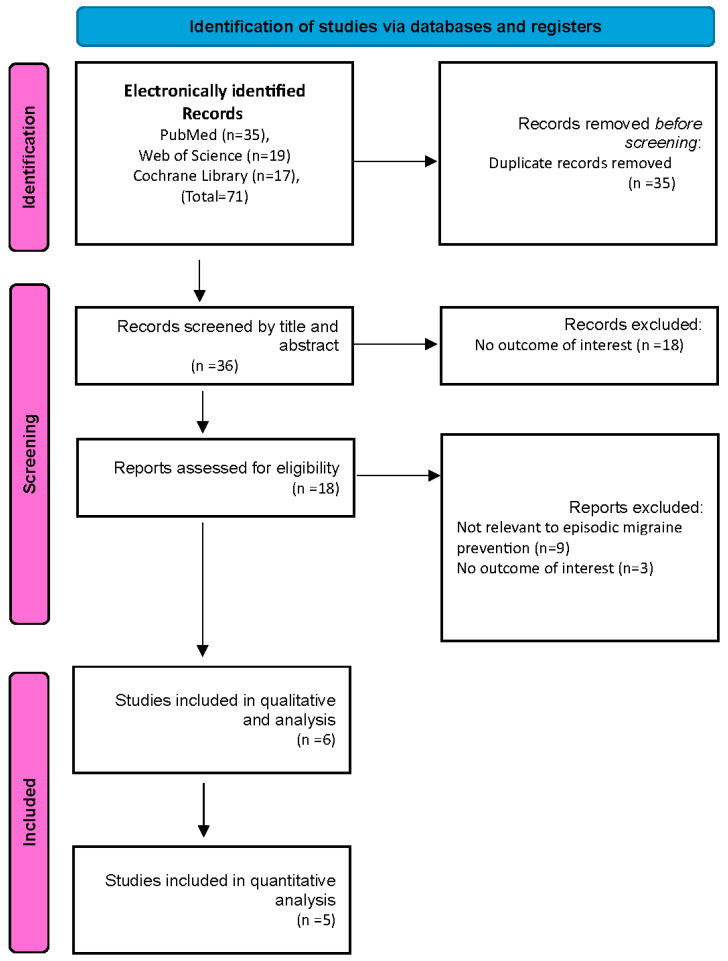
PRISMA flowchart.

**Figure 2 jcm-13-06713-f002:**
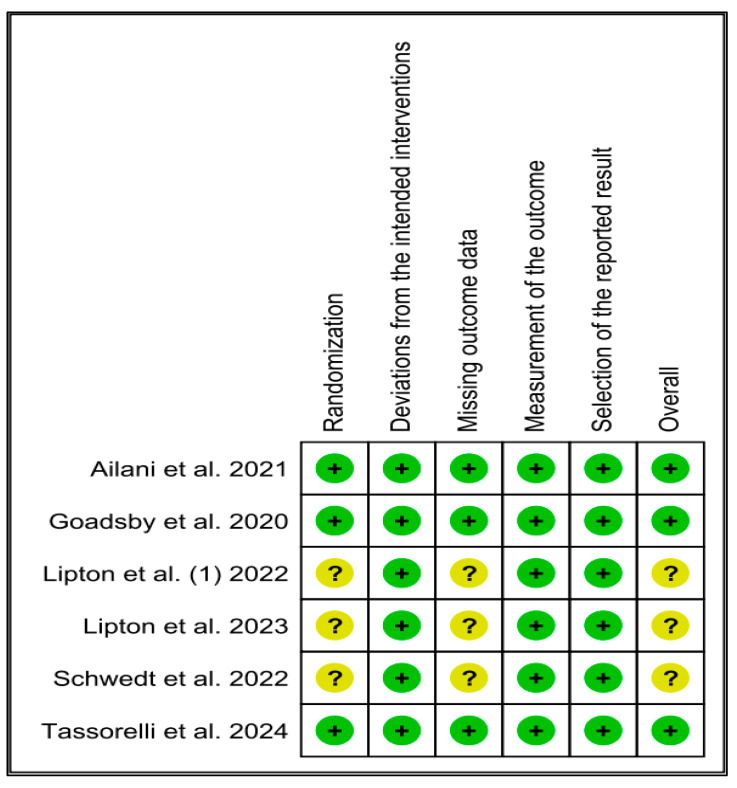
Quality assessment of the included studies [9,11,25,26,27,28].

**Figure 3 jcm-13-06713-f003:**
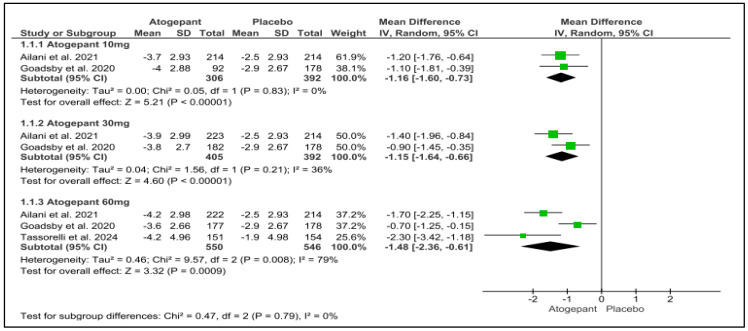
Mean monthly migraine days change from baseline forest plot [11,26,28].

**Figure 4 jcm-13-06713-f004:**
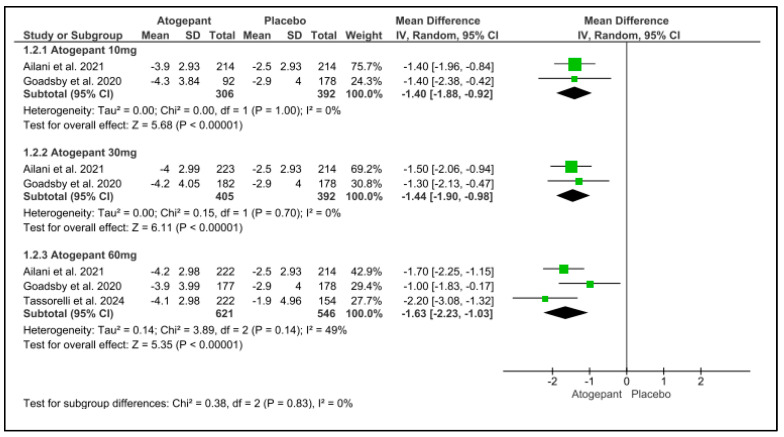
Mean monthly headache days change from baseline forest plot [11,26,28].

**Figure 5 jcm-13-06713-f005:**
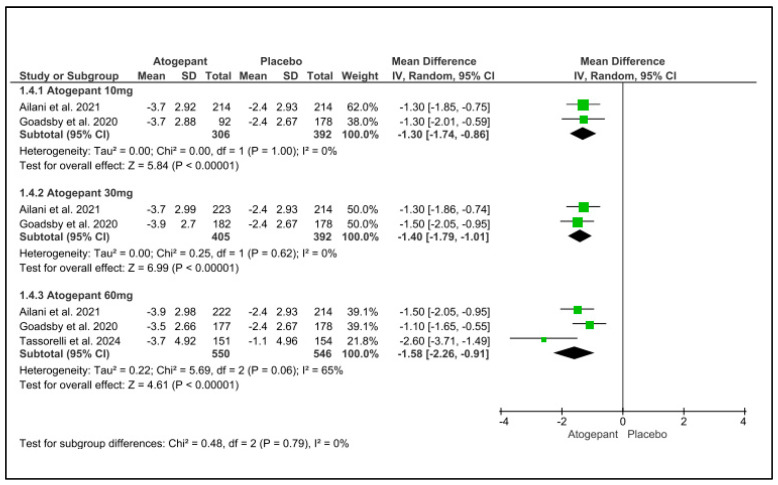
Acute-medications-use days change from baseline forest plot [11,26,28].

**Figure 6 jcm-13-06713-f006:**
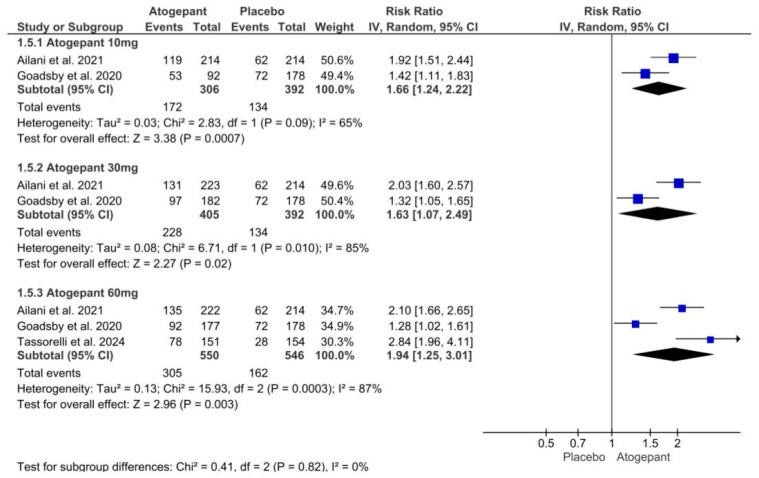
≥50% reduction in monthly migraine days forest plot [11,26,28].

**Figure 7 jcm-13-06713-f007:**
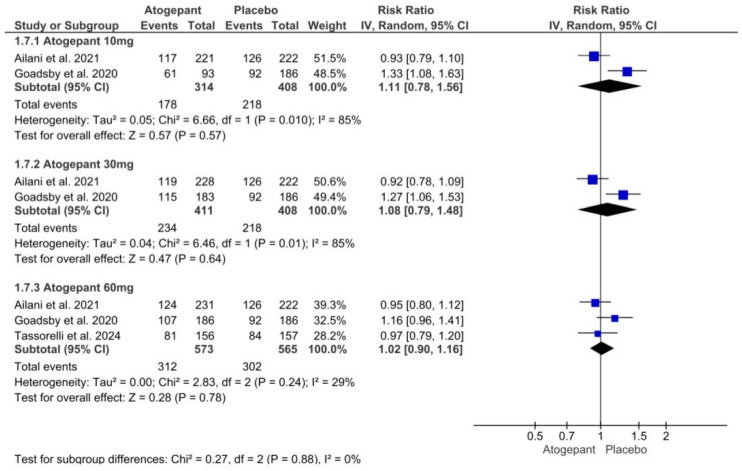
Any adverse events forest plot [11,26,28].

**Figure 8 jcm-13-06713-f008:**
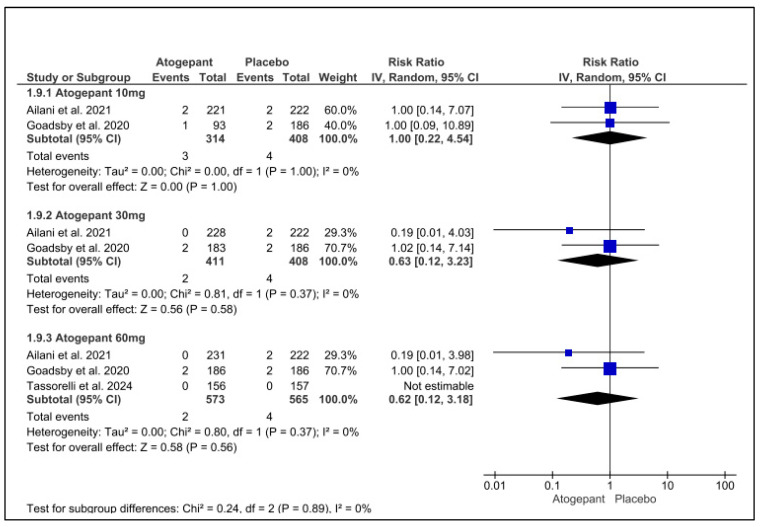
Serious adverse events forest plot [11,26,28].

**Figure 9 jcm-13-06713-f009:**
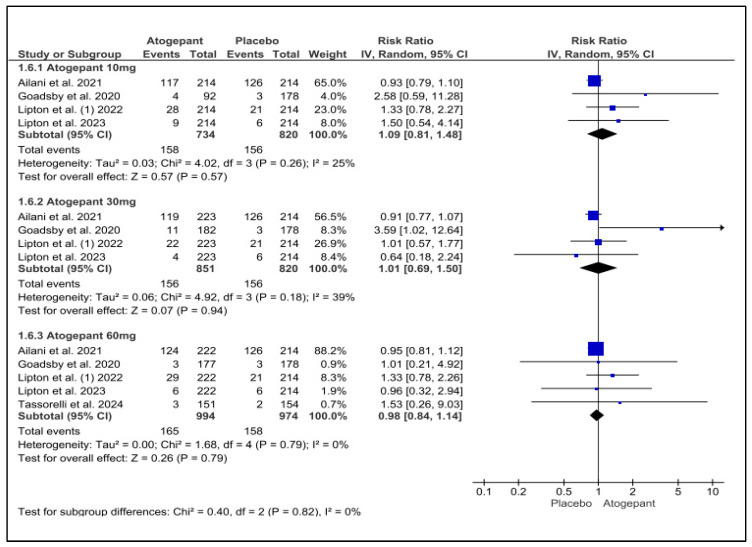
Discontinuation due to adverse events forest plot [9,11,26,27,28].

**Table 1 jcm-13-06713-t001:** Baseline characteristics of the Included Studies.

Variable	Goadsby et al., 2020 [26]				Schwedt et al., 2022 [25]			
	Placebo	Atogepant 10 mg Once Daily	Atogepant 30 mg Once Daily	Atogepant 60 mg Once Daily	Placebo	Atogepant 10 mg Once Daily	Atogepant 30 mg Once Daily	Atogepant 60 mg Once Daily
Total sample size	186	93	183	186	222	221	228	231
Modified intention to treat population	178	92	182	177	214	214	223	222
Age, years	40.5 (11.7)	39.4 (12.4)	41.0 (13.6)	40.4 (11.7)	40.3 (12.8)	41.4 (12.1)	42.1 (11.7)	42.5 (12.4)
Female sex	154 (83%)	82 (88%)	166 (91%)	156 (84%)	198 (89.2%)	200 (90.5%)	204 (89.5%)	199 (86.1%)
White race	137 (74%)	69 (74%)	145 (79%)	133 (72%)	194 (87.4%)	181 (81.9%)	185 (81.1%)	192 (83.1%)
Black race	45 (24%)	20 (22%)	29 (16%)	44 (24%)	N/A	N/A	N/A	N/A
Other races	4 (2%)	4 (4%)	9 (5%)	9 (5%)	199 (89.6%)	200 (90.5%)	209 (91.7%)	217 (93.9%)
Body mass index, kg/m	30.4 (7.6)	29.9 (7.3)	30.0 (7.1)	30.0 (7.8)	30.8 (8.7)	30.4 (7.6)	31.2 (7.6)	29.9 (7.3)
Monthly migraine days	7.8 (2.5)	7.6 (2.5)	7.6 (2.4)	7.7 (2.6)	7.5 (2.4)	7.5 (2.5)	7.9 (2.3)	7.8 (2.3)
Migraine with aura	45 (24%)	21 (23%)	37 (20%)	36 (19%)	N/A	N/A	N/A	N/A
Migraine without aura	94 (51%)	48 (52%)	93 (51%)	96 (52%)	N/A	N/A	N/A	N/A
Monthly headache days	9.1 (2.7)	8.9 (2.7)	8.7 (2.5)	8.9 (2.8)	8.4 (2.6)	8.4 (2.8)	8.8 (2.6)	9.0 (2.6)
Monthly acute medication use days	6.6 (3.2)	6.2 (3.3)	6.6 (3.0)	6.8 (3.3)	6.5 (3.2)	6.6 (3.0)	6.7 (3.0)	6.9 (3.2)
**Variable**	**Tassorelli et al., 2024** [28]				**Ailani et al., 2021** [11]			
	**Placebo**	**Atogepant** **10 mg** **Once Daily**	**Atogepant 30 mg** **Once Daily**	**Atogepant 60 mg** **Once Daily**	**Placebo**	**Atogepant** **10 mg** **Once Daily**	**Atogepant** **30 mg** **Once Daily**	**Atogepant** **60 mg** **Once Daily**
Total sample size	157	N/A	N/A	156	222	221	228	231
Modified intention to treat population	154	N/A	N/A	151	214	214	223	222
Age, years	43.4 (10.3)	N/A	N/A	40.9 (10.7)	40.3 (12.8)	41.4 (12.0)	42.1 (11.7)	42.5 (12.4)
Female sex	141 (90%)	N/A	N/A	139 (89%)	198 (89.2%)	200 (90.5%)	204 (89.5%)	199 (86.1%)
White race	151 (96%)	N/A	N/A	149 (96%)	194 (87.4%)	181 (81.9%)	185 (81.1%)	192 (83.1%)
Black race	4 (3%)	N/A	N/A	3 (2%)	24 (10.8%)	34 (15.4%)	38 (16.7%)	28 (12.1%)
Other races	2 (1%)	N/A	N/A	2 (1%)	4 (1.8%)	6 (2.8%)	5 (2.1%)	10 (4.3%)
Body mass index, kg/m	26.2 (5.2)	N/A	N/A	25.6 (4.9)	30.8 (8.7)	30.3 (7.6)	31.1 (7.6)	29.9 (7.3)
Monthly migraine days	N/A	N/A	N/A	N/A	7.7 (2.6)	7.2 (2.5)	7.3 (2.4)	7.3 (2.4)
Migraine with aura	N/A	N/A	N/A	N/A	N/A	N/A	N/A	N/A
Migraine without aura	N/A	N/A	N/A	N/A	N/A	N/A	N/A	N/A
Monthly headache days	N/A	N/A	N/A	N/A	9.5 (2.8)	9.3 (2.7)	9.2 (2.7)	9.1 (2.7)
Monthly acute medication use days	N/A	N/A	N/A	N/A	N/A	N/A	N/A	N/A
**Variable**	**Lipton et al., 2023** [27]				**Lipton et al., 2022** [9]			
	**Placebo**	**Atogepant 10 mg** **Once Daily**	**Atogepant 30 mg** **Once Daily**	**Atogepant 60 mg** **Once Daily**	**Placebo**	**Atogepant** **10 mg** **Once Daily**	**Atogepant** **30 mg** **Once Daily**	**Atogepant** **60 mg** **Once Daily**
Total sample size	222	221	228	231	222	221	228	231
Modified intention to treat population	214	214	223	222	214	214	223	222
Age, years	40.3 (12.9)	41.5 (12.0)	42.2 (11.7)	42.8 (12.3)	40.3 (12.8)	41.4 (12.1)	42.1 (11.7)	42.5 (12.4)
Female sex	190 (88.8%)	193 (90.2%)	199 (89.2%)	191 (86.0%)	198 (89.2%)	200 (90.5%)	204 (89.5%)	199 (86.1%)
White race	188 (87.9%)	176 (82.2%)	181 (81.2%)	184 (82.9%)	194 (87.4%)	181 (81.9%)	185 (81.1%)	192 (83.1%)
Black race	22 (10.3%)	32 (15.0%)	37 (16.6%)	27 (12.2%)	24 (10.8%)	34 (15.4%)	38 (16.7%)	28 (12.1%)
Other races	4 (1.8%)	6 (2.8%)	5 (2.2%)	11 (5%)	4 (1.8%)	6 (2.8%)	5 (2.1%)	11 (4.7%)
Body mass index, kg/m	N/A	N/A	N/A	N/A	30.8 (8.7%)	30.4 (7.6%)	31.2 (7.6%)	29.9 (7.3%)
Monthly migraine days baseline	7.5 (2.4)	7.5 (2.5)	7.9 (2.3)	7.8 (2.3)	7.5 (2.4)	7.5 (2.5)	7.9 (2.3)	7.8 (2.3)
Migraine with aura	N/A	N/A	N/A	N/A	N/A	N/A	N/A	N/A
Migraine without aura	N/A	N/A	N/A	N/A	N/A	N/A	N/A	N/A
Monthly headache days	8.4 (2.6)	8.4 (2.8)	8.8 (2.6)	9.0 (2.6)	N/A	N/A	N/A	N/A
Monthly acute medication use days	6.5 (3.2)	6.6 (3.0)	6.7 (3.0)	6.9 (3.2)	N/A	N/A	N/A	N/A

Data are reported as: N (%) or mean (SD); N/A: not applicable.

**Table 2 jcm-13-06713-t002:** Summary of the reported treatment-related treatment emergent adverse events (TEAE).

	Study Reference	Placebo	Atogepant 10 mg Once Daily	Atogepant 30 mg Once Daily	Atogepant 60 mg Once Daily
Any treatment-related TEAE	Lipton et al.’s (2022) [9]	20/222 (9%)	51/221 (23.1%)	34/228 (14.9%)	45/231 (19.5%)
Constipation		1/222 (5%)	17/221 (7.7%)	16/228 (7%)	16/231 (6.9%)
Nausea		4/222 (1.8%)	11/221 (5%)	10/228 (4.4%)	14/231 (6.1%)
Any treatment-related TEAE	Ailani et al. (2021) [11]	20/222 (9%)	51/221 (23.1%)	34/228 (14.9%)	45/231 (19.5%)
Constipation		1/222 (0.5%)	17/221 (7.7%)	16/228 (7%)	16/231 (6.9%)
Upper respiratory tract infection		10/222 (4.5%)	9/221 (4.1%)	13/228 (5.7%)	9/231 (3.9%)
Nausea		4/222 (1.8%)	11/221 (5%)	10/228 (4.4%)	14/231 (6.1%)
Any treatment-related TEAE	Schwedt et al. (2022) [25]	56.80%	N/A	N/A	N/A
Nausea		N/A	N/A	N/A	N/A
Constipation		N/A	N/A	N/A	N/A
Fatigue		N/A	N/A	N/A	N/A
Any treatment-related TEAE	Goadsby et al. (2020) [26]	30/186 (16%)	17/93 (18%)	39/183 (21%)	42/186 (23%)
Nausea		5/186 (3%)	3/93 (3%)	10/183 (5%)	11/186 (6%)
Constipation		2/186 (1%)	1/93 (1%)	10/183 (5%)	8/186 (4%)
Fatigue		4/186 (2%)	1/93 (1%)	2/183 (1%)	4/186 (2%)
Any treatment-related TEAE	Lipton et al. (2023) [27]	N/A	N/A	N/A	N/A
Nausea		N/A	N/A	N/A	N/A
Constipation		N/A	N/A	N/A	N/A
Fatigue		N/A	N/A	N/A	N/A
Any treatment-related TEAE	Tassorelli et al. (2024) [28]	14/157 (9%)	N/A	N/A	31/156 (20%)
Constipation		3/157 (2%)	N/A	N/A	13/156 (8%)
Nausea		3/157 (2%)	N/A	N/A	8/156 (5%)
Decreased appetite		0	N/A	N/A	5/156 (3%)

Date was reported as: N (%), N/A: not applicable.

## Data Availability

The original contributions presented in the study are included in the article. Further inquiries can be directed to the corresponding author.

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
