# Peer review of "Safety and Efficacy of Atogepant for the Preventive Treatment of Migraines in Adults: A Systematic Review and Meta-Analysis"

_jcm, 2024, doi:10.3390/jcm13226713_

Round 1
Reviewer 1 Report
Comments and Suggestions for Authors
This manuscript entitled 'Safety and Efficacy of Atogepant for the Preventive Treatment of Migraines in Adults: A Systematic Review and Meta-Analysis' is well-organized and fluency written. However, the manuscript in its current version requires some some revisions:
Introduction
This manuscript discusses the role of atogepant in the preventive treatment of episodic migraines. It would be helpful to elaborate on the advantages of atogepant over other CGRP receptor antagonists, such as rimegepant. Explain it briefly.
Methods
A summary table showing the risk of bias assessment for each included study is required. "Low quality or high bias risk" in the exclusion criteria need to be more clearly explained.
The basis for choosing a fixed-effect or random-effect model, and the content of sensitivity analysis need to be explained in more detail.
Results
1. Please provide a table on the demographic and clinical characteristics of the 4598 participants at baseline, or characteristics of the included studies.
2. Regarding the presentation of statistical significance, if the P-value is less than 0.001, it should be recorded as p<0.001.
3. In Section 3.5, you report a significant reduction in mean monthly migraine days for the 60 mg dose of atogepant. However, you also mention a substantial degree of heterogeneity for this subgroup. Please discuss the potential sources of this heterogeneity and how it might affect the interpretation of your results.
4. For the adverse events reported in Section 3.9, a summary table comparing the incidence of adverse events across different doses of atogepant and the placebo group will be helpful.
Discussion
You mentioned 60 mg dose showing the greatest reduction. Please discuss any variations observed in the relief provided by different doses (10mg ,30mg and 60mg) of atogepant.
Please ensure that all figures and tables mentioned in the text are included and properly labeled in the manuscript.
Author Response
Comment 1: [This manuscript discusses the role of atogepant in the preventive treatment of episodic migraines. It would be helpful to elaborate on the advantages of atogepant over other CGRP receptor antagonists, such as rimegepant. Explain it briefly.]
Response 1: [Thank you for your insightful feedback. We have mentioned the role of atogepant over the CGRP pathway. Please refer to page 2, last paragraph, lines 88-98.]
Comment 2: [A summary table showing the risk of bias assessment for each included study is required. "Low quality or high bias risk" in the exclusion criteria need to be more clearly explained.
The basis for choosing a fixed-effect or random-effect model, and the content of sensitivity analysis need to be explained in more detail.]
Response 2: [Thank you for your insightful feedback. We have corrected the exclusion criteria. Due to the different demographic characteristics, we implemented a random-effects model. Details are provided on page 3, paragraph 3, lines 122-123, and page 4, paragraph 2, lines 157-161.]
Comment 3: [
- Please provide a table on the demographic and clinical characteristics of the 4598 participants at baseline, or characteristics of the included studies.
- Regarding the presentation of statistical significance, if theP-value is less than 0.001, it should be recorded as p<0.001.
- In Section 3.5, you report a significant reduction in mean monthly migraine days for the 60 mg dose of atogepant. However, you also mention a substantial degree of heterogeneity for this subgroup.Please discuss the potential sources of this heterogeneity and how it might affect the interpretation of your results.
- For the adverse events reported in Section 3.9, a summary table comparing the incidence of adverse events across different doses of atogepant and the placebo groupwill be helpful.]
Response 3: [1.Thank you for your insightful feedback. Kindly refer to the attached Table 1, located on pages 7-9, lines 250-257.
2. Thank you for your insightful feedback. Kindly note that these values have been corrected throughout the manuscript.
3. Thank you for your insightful feedback. Kindly note that, according to our statistical analysis, the methodology and demographics were nearly similar, suggesting that the heterogeneity may be attributed to random error. We cannot speculate on the effect of this heterogeneity, as it is likely due to random variation, page 9, last paragraph, line 277.
4. Thank you for your insightful feedback. We have included a table comparing these adverse events; please refer to Table 2 on pages 13.]
Comment 4: [You mentioned 60 mg dose showing the greatest reduction. Please discuss any variations observed in the relief provided by different doses (10mg ,30mg and 60mg) of atogepant.]
Response 4: [Thank you for your insightful feedback. We have noted the difference between the atogepant dosages; please refer to page 15-16, paragraph 2, lines 406-408.]
Comment 5: [Please ensure that all figures and tables mentioned in the text are included and properly labeled in the manuscript.]
Response 5: [Thank you for your insightful feedback. We have ensured the incorporation of all tables and figures.]
Reviewer 2 Report
Comments and Suggestions for Authors
The manuscript titled "Safety and Efficacy of Atogepant for the Preventive Treatment of Migraines in Adults: A Systematic Review and Meta-Analysis" presents a thorough investigation into the efficacy and safety of atogepant, a CGRP receptor antagonist, for preventing episodic migraines in adults. This systematic review and meta-analysis focused on randomized, double-blind, placebo-controlled trials, with primary outcomes including changes in mean monthly migraine days (MMDs) and mean headache days (MHDs) over 12 weeks. Results indicate that atogepant significantly reduced MMDs and MHDs across all doses (10 mg, 30 mg, 60 mg), with the 60 mg dose demonstrating the most substantial reduction. Additionally, atogepant decreased acute medication usage and improved 50% responder rates. However, several important issues need further consideration.
1. Although the study shows efficacy at different doses, a detailed dose-response analysis could provide insights into optimal dosing for different patient populations. This could help in personalized treatment planning.
2. There is insufficiency on adverse events data. The reporting on adverse events could be more detailed to better understand the safety profile, especially at higher doses.
3. The study does not address any aspect on the potential for personalized treatment strategies based on patient-specific factors.
4. The discussion could benefit from a more detailed comparison of the findings with those of other relevant studies. While the study mentions that atogepant has a favorable safety profile, a deeper analysis of how these results align with or differ from other CGRP receptor antagonists would be valuable.
5. The study notes an increase in adverse events with higher doses of atogepant but it does not adequately address the implications of this dose-response relationship.. This is particularly important for balancing the benefits of treatment with the risk of side effects.
6. The absence of long-term safety data is a critical gap, especially for a treatment intended for chronic use. While the study briefly mentions the limitations, the short follow-up period of 12 weeks may not be sufficient to assess the long-term efficacy and safety of atogepant. Many chronic conditions like migraine require longer-term data to understand the sustainability of treatment effects and the potential for late-onset adverse events. A more thorough discussion on the potential impact of these limitations on the results would be beneficial.
Author Response
Comment 1: [Although the study shows efficacy at different doses, a detailed dose-response analysis could provide insights into optimal dosing for different patient populations. This could help in personalized treatment planning.]
Response 1: [Thank you for your insightful feedback. We could not perform a meta-analysis on dose-response analysis due to insufficient data. We have incorporated a narrative paragraph about the dose-response analysis on page 17, paragraph 3, lines 481-489]
Comment 2: [There is insufficiency on adverse events data. The reporting on adverse events could be more detailed to better understand the safety profile, especially at higher doses.]
Response 2: [Thank you for your insightful feedback. We could not perform a detailed dose-response analysis due to insufficient data. However, we have mentioned the adverse events associated with different available dosages and incorporated Table 2 in Section 3.9, pages 12-13, lines 321-360]
Comment 3: [The study does not address any aspect on the potential for personalized treatment strategies based on patient-specific factors.]
Response 3: [Thank you for your insightful feedback. These issues have been addressed. Please refer to page 17, paragraph 3, lines 481-489]
Comment 4: [The discussion could benefit from a more detailed comparison of the findings with those of other relevant studies. While the study mentions that atogepant has a favorable safety profile, a deeper analysis of how these results align with or differ from other CGRP receptor antagonists would be valuable.]
Response 4: [Thank you for your insightful feedback. These issues have been addressed. Please refer to pages 17, first paragraph, lines 462-471]
Comment 5: [The study notes an increase in adverse events with higher doses of atogepant but it does not adequately address the implications of this dose-response relationship.. This is particularly important for balancing the benefits of treatment with the risk of side effects.]
Response 5: [Thank you for your insightful feedback. We could not perform a detailed dose-response analysis due to insufficient data. However, we have mentioned the adverse events associated with different available dosages and incorporated Table 2 in Section 3.9, pages 11-13, lines 321-360]
Comment 6: [The absence of long-term safety data is a critical gap, especially for a treatment intended for chronic use. While the study briefly mentions the limitations, the short follow-up period of 12 weeks may not be sufficient to assess the long-term efficacy and safety of atogepant. Many chronic conditions like migraine require longer-term data to understand the sustainability of treatment effects and the potential for late-onset adverse events. A more thorough discussion on the potential impact of these limitations on the results would be beneficial.]
Response 6: [Thank you for your insightful feedback. These issues have been addressed. Please refer to page 18, first paragraph, lines 521-529]
Reviewer 3 Report
Comments and Suggestions for Authors
Comments to authors:
This study by Alrasheed et al. investigated the efficacy and safety of atogepant in preventing episodic migraine in adults via a systemic review and meta-analysis approach. The systemic review and meta-analysis followed the Preferred Reporting Items for Systematic Reviews and Meta-Analyses (PRISMA) guidelines, and six randomized control trials with ~4,600 patients were included in the final meta-analysis studies. The results demonstrated atogepant efficacy and safety at 3 dose levels (10, 30 and 60 mg) in mean monthly migraine and headache days, acute medication use days, outcome of a 50% reduction, adverse events, serious adverse events, and discontinuation due to adverse events. The conclusions were appropriate based on the analysis results and provide further evidence on the efficacy and safety of atogepant for episodic migraine prophylaxis in adults. Nevertheless, there have been other recently published meta-analyses on the same or similar research questions and thus undermining the novelty of the current study. My detailed comments are below:
Main concerns/comments:
1. Three out of the six studies included in the final meta-analysis showed overall concerns and specifically concerns in randomization and missing outcome data (Figure 1). In all of the efficacy and safety outcomes, the Schwedt et al. 2022 study was not included in the meta-analysis at all, and only the “discontinuation due to adverse events” outcome used data from the rest of the 5 studies, while for the rest of the outcome analysis, only the 3 studies without bias were included. Therefore, it is misleading to claim 6 RCTs and 4598 participants were included in the analysis.
2. Also, please elaborate on where those concerns lie for Lipton et al. (1) 2022, Lipton et al. 2023, and Schwedt et al. 2022, and discuss why the three studies were excluded from analyses of all outcomes but one (Discontinuation Due to Adverse Events)
3. The authors discussed the difference between this current study to several previous meta-analyses on the same or similar question, including Tao et al 2022 (reference 12 in the manuscript), Lattanzi et al. 2022 (reference 16), Lopez, L.M. et al. 2024 (reference 35), and Hou et al. 2024. Please provide a table to compare the differences and similarities of these published meta-analyses.
4. Please provide a table comparing the six studies used in the current meta-analysis, including the number of patients, study design, demographic information, et cetera, at each dose level and placebo groups. This will provide more insight into the variations seen consistently at the 60 mg dose group.
5. The eligibility criteria stated that “participants needed to experience 4–14 monthly migraine days (MMDs) in the 3 months before screening and record this data in an electronic diary during a 28-day baseline period”, while the diagnostic criteria for episodic migraine is headache attacks occurring on fewer than 15 days per month. Please comment on and reconcile these discrepancies.
Minor points:
1. There are many sentences in the introduction section without any citations, including but not limited to “Migraine can be classified into two major types according to the International Classification of Headache Disorders (ICHD-3): episodic and chronic.”, “CGRP plays a crucial role in migraine pathophysiology by modulating pain pathways and vascular functions”, “For the prevention of migraine, there are four monoclonal antibodies that target the CGRP receptor (erenumab) or ligand (galcanezumab, fremanezumab, and eptinezumab). Rimegepant and ubrogepant, two oral CGRP receptor antagonists, are licensed for the management of migraine attacks”. Please add these references assuming the audience is not familiar with migraine and its treatment.
2. The text in Figure 2,6,7 looks distorted. Please scale the figures instead of only changing the width or length.
3. As common as the use of NSAIDs is, please add the full name before using the abbreviation NSAIDs.
4. This sentence is confusing: “Although, these were traditional treatments for migraine, they are non-specific migraine treatments with limited degrees of effectiveness, ……”. The authors might have intended to say “however” instead of “although”. Please confirm and revise for clarity.
5. In this sentence “The development of CGRP antagonists offers a targeted approach to migraine treatment, providing relief with fewer side effects compared to traditional therapies [15]”, Reference 15 does not mention CGRP or CGRP antagonists at all. Please replace the reference here.
Author Response
Comment 1: [Three out of the six studies included in the final meta-analysis showed overall concerns and specifically concerns in randomization and missing outcome data (Figure 1). In all of the efficacy and safety outcomes, the Schwedt et al. 2022 study was not included in the meta-analysis at all, and only the “discontinuation due to adverse events” outcome used data from the rest of the 5 studies, while for the rest of the outcome analysis, only the 3 studies without bias were included. Therefore, it is misleading to claim 6 RCTs and 4598 participants were included in the analysis.]
Response 1: [Thank you for your insightful feedback. Please note that we followed PRISMA guidelines in conducting the current study. Six trials were included in the narrative review, and five in the meta-analysis, depending on data availability. Please refer to the details of inclusion in the PRISMA flowchart, Figure 1, on page 5]
Comment 2: [Also, please elaborate on where those concerns lie for Lipton et al. (1) 2022, Lipton et al. 2023, and Schwedt et al. 2022, and discuss why the three studies were excluded from analyses of all outcomes but one (Discontinuation Due to Adverse Events)]
Response 2: [Thank you for your insightful feedback. Please refer to the reasons for concerns in Figure 2 on page 6. The inclusion was based on data availability, as these studies did not document all of the outcome measures data]
Comment 3: [The authors discussed the difference between this current study to several previous meta-analyses on the same or similar question, including Tao et al 2022 (reference 12 in the manuscript), Lattanzi et al. 2022 (reference 16), Lopez, L.M. et al. 2024 (reference 35), and Hou et al. 2024. Please provide a table to compare the differences and similarities of these published meta-analyses.]
Response 3: [Thank you for your insightful feedback. Please note that we have discussed the similarities between recent studies and ours, as well as the substantial differences, in accordance with the PRISMA guidelines and checklist that was sent to the editor. Please refer to page 17-18, lines 490-517]
Comment 4: [Please provide a table comparing the six studies used in the current meta-analysis, including the number of patients, study design, demographic information, et cetera, at each dose level and placebo groups. This will provide more insight into the variations seen consistently at the 60 mg dose group ]
Response 4: [Thank you for your insightful feedback. Please refer to Table 1 on pages 7-9]
Comment 5: [The eligibility criteria stated that “participants needed to experience 4–14 monthly migraine days (MMDs) in the 3 months before screening and record this data in an electronic diary during a 28-day baseline period”, while the diagnostic criteria for episodic migraine is headache attacks occurring on fewer than 15 days per month. Please comment on and reconcile these discrepancies]
Response 5: [Thank you for your insightful feedback. This criterion was based on the available clinical trials, as 4-4 days may justify further investigation and treatment]
Comment 6: [
There are many sentences in the introduction section without any citations, including but not limited to “Migraine can be classified into two major types according to the International Classification of Headache Disorders (ICHD-3): episodic and chronic.”, “CGRP plays a crucial role in migraine pathophysiology by modulating pain pathways and vascular functions”, “For the prevention of migraine, there are four monoclonal antibodies that target the CGRP receptor (erenumab) or ligand (galcanezumab, fremanezumab, and eptinezumab). Rimegepant and ubrogepant, two oral CGRP receptor antagonists, are licensed for the management of migraine attacks”. Please add these references assuming the audience is not familiar with migraine and its treatment.]
Response 6: [Thank you for your insightful feedback. Please note that we have followed the journal's citation and referencing styles. Continuous data from the same references are cited at the end of the paragraph, as is the case in the mentioned instances]
Comment 7: [The text in Figure 2,6,7 looks distorted. Please scale the figures instead of only changing the width or length.]
Response 7: [Thank you for your insightful feedback. Please refer to the same figures where they are scaled up. The figures cannot be manipulated, as they are derived directly from the statistical software]
Comment 8: [As common as the use of NSAIDs is, please add the full name before using the abbreviation NSAIDs.]
Response 8: [Thank you for your insightful feedback. Please refer to page 2, paragraph 2, line 58]
Comment 9: [sentence is confusing: “Although, these were traditional treatments for migraine, they are non-specific migraine treatments with limited degrees of effectiveness, ……”. The authors might have intended to say “however” instead of “although”. Please confirm and revise for clarity]
Response 9: [Thank you for your insightful feedback. The word has been changed and corrected. Please refer to page 2, paragraph 2, line 64.]
Comment 10: [In this sentence “The development of CGRP antagonists offers a targeted approach to migraine treatment, providing relief with fewer side effects compared to traditional therapies [15]”, Reference 15 does not mention CGRP or CGRP antagonists at all. Please replace the reference here.]
Response 10: [Thank you for your insightful feedback. The reference has been corrected. Please refer to page 2, paragraph 3, lines 71-73]
Round 2
Reviewer 1 Report
Comments and Suggestions for Authors
I have reviewed the revised manuscript and find that the authors have adequately addressed my previous concerns. The revisions have improved the quality of the paper
Reviewer 3 Report
Comments and Suggestions for Authors
No additional comments.